# Genetic Hearing Loss Affects Cochlear Processing

**DOI:** 10.3390/genes13111923

**Published:** 2022-10-22

**Authors:** Cris Lanting, Ad Snik, Joop Leijendeckers, Arjan Bosman, Ronald Pennings

**Affiliations:** Department of Otorhinolaryngology, Radboud University Medical Centre, Donders Institute for Brain, Cognition and Behaviour, Postbus 9101, 6500 HB Nijmegen, The Netherlands

**Keywords:** speech-in-noise, loudness growth, gap detection, frequency discrimination, otogenetics, hereditary hearing loss

## Abstract

The relationship between speech recognition and hereditary hearing loss is not straightforward. Underlying genetic defects might determine an impaired cochlear processing of sound. We obtained data from nine groups of patients with a specific type of genetic hearing loss. For each group, the affected cochlear site-of-lesion was determined based on previously published animal studies. Retrospectively obtained speech recognition scores in noise were related to several aspects of supra-threshold cochlear processing as assessed by psychophysical measurements. The differences in speech perception in noise between these patient groups could be explained by these factors and partially by the hypothesized affected structure of the cochlea, suggesting that speech recognition in noise was associated with a genetics-related malfunctioning of the cochlea. In particular, regression models indicate that loudness growth and spectral resolution best describe the cochlear distortions and are thus a good biomarker for speech understanding in noise.

## 1. Introduction

The relationship between speech recognition and the degree of sensorineural hearing loss is not straightforward. Patients, for example, with an average sensorineural hearing loss of 70 dB HL and an adequate amplification may have speech recognition scores that vary between 10 and 80% [1]. Similarly, patients with an equal hearing loss, differ greatly in their ability to understand speech-in-noise [2,3,4,5]. Ignoring a central neural processing deficit and other top-down influences such as cognitive factors as causes [6,7,8,9], the poor relationship between speech recognition and hearing impairment is supposedly related to variable degrees of the deficient processing of speech by an impaired cochlea [10,11].

While hearing loss does explain some of the difficulties patients report, it has become clear that it cannot be the only factor. This is, for example, illustrated by patients with autosomal, a dominantly inherited form of hereditary hearing loss type 2 and 9 (DFNA2 and DFNA9), who have comparable high-frequency hearing losses at a relative young age [12]. DFNA9 patients with a pure tone average (PTA_1,2,4 kHz_) of 90 dB HL had an average phoneme score of 40%, whereas this percentage for DFNA2 patients with the same degree of hearing loss was approximately 80%. Speech recognition scores in silence and in noise thus seem to be rather uniquely related to the underlying genetic type of hearing impairment [13]. 

In line with these findings, we hypothesize that variations in speech recognition between patients is not primarily related to the degree of hearing impairment but also the degree of impaired cochlear processing. The latter mainly depends on which part of the cochlea is affected, e.g., hair cells responsible for mechanotransduction [14,15], the stria vascularis and thus the endocochlear potential [16], and the tectorial membrane and the mechanical properties of the organ of Corti [17]. Schuknecht et al. found that the shape of the audiogram in age-related hearing loss mainly related to atrophy of the stria vascularis based on cell counts of various cell types in histological samples of the inner ear [18]. However, more recent work using more sensitive methods indicates that age-related hearing loss is primarily due to hair cell pathology instead of strial atrophy [19,20]. In data where word-recognition scores were available, auditory nerve fiber (ANF) survival was predictive of word-recognition scores, even when the PTA was included in the analysis [21]. 

These new insights highlight that the site of pathology, or cochlear site-of-lesion, may ultimately determine the thresholds and the ability to understand speech in quiet and speech-in-noise. In this work, we further highlight this by studying specific groups of patients. Over the last decade, we published the results on psychophysical and speech-in-noise tests obtained in nine different groups of hearing-impaired patients with a particular type of genetic hearing impairment [13,22,23,24,25,26,27,28,29] (see also Table 1). Data from these publications were used to test the hypothesis that the impairment in speech-in-noise is not primarily related to the degree of hearing impairment but more to the degree of impaired cochlear processing. Furthermore, results from this study may relate to biomarkers of auditory function that comprise a deep auditory phenotype that is essential for patient selection and for evaluation of inner ear therapeutic studies. Moreover, it may be used to further develop tests that characterize cochlear function and processing more precisely than the current practice of pure tone audiometry and speech understanding.

## 2. Patients and Methods

In 2004, De Leenheer et al. introduced a test battery consisting of psychophysical (loudness perception, temporal and spectral processing) and speech recognition in noise tests to assess cochlear processing [22]. Over time, these tests have been used in nine different groups of patients with a specific type of genetic hearing loss (Table 1). Additionally, in some of these studies, results have also been collected for normal hearing controls. The results from these previous studies were used in the present study to test whether variations in speech recognition in noise between patients is related to the degree of impaired cochlear processing likely caused by the underlying genetic disorder.

The test battery consisted of four tests performed in a standardized way in all patient groups. *Loudness growth* was measured with 0.5 kHz and 2 kHz tones. The best-fit curve through the loudness growth data was calculated, and its slope was the primary outcome measure (Slope of the Loudness Growth, SLG). It was decided to present the slope relative to (divided by) the slope of normal hearing subjects. This means that a relative slope of 1 reflects a loudness growth similar to normal hearing subjects. In the case of loudness recruitment, the relative slope is larger than 1 [30]. 

*Gap detection (GDT),* or the shortest perceived period of silence between two noise bursts, was measured using band-filtered white noise with center frequencies of 0.5 kHz and 2 kHz. The smallest detectable gap is presented relative to (divided by) the norm value to obtain a relative measure. A value of 1 means that the smallest gap detected is not different from the norm, and a value larger than 1 indicates reduced temporal precision.

*The difference limen for frequency (DLF)* was measured at 0.5 kHz and 2 kHz with frequency-modulated tones. The lowest modulation frequency that the patient detected was taken as the DLF, and it is presented relative to (divided by) the norm value. 

The critical signal-to-noise ratio was measured using the *speech recognition in noise test*, also referred to as the Plomp-test [10,11]. In this test, the speech reception threshold (SRT) is determined with adaptive trial-by-trial level adjustments of the speech compared to the noise. The SRT estimates the speech level (or the speech-to-noise ratio, SNR) at which 50% of the sentences are correctly recognized. This SRT comprises two effects: the audibility or attenuation (A-factor), referring to the increased threshold determined by pure tone audiometry. The other factor, coined as the cochlear distortion factor (CDF or D-factor), relates to the increased or added SNR needed for recognition. The CDF is obtained by taking the SRT and subtracting the norm-value. If CDF = 0, then the SRT-in-noise is normal. If CDF > 0, the patient has more difficulty understanding speech-in-noise than the controls. Often this term refers to suprathreshold deficits, such as reduced frequency selectivity or impaired temporal processing. 

However, as pointed out by Houtgast and Festen (Houtgast and Festen 2008), the D-factor [10] may also depend on the shape of the audiogram and, thus, the audibility. For the Usher syndrome type 2a (USH2a) group, the DFNA10 group, the Non-Ocular (NO) Stickler syndrome group, and the HDR (Hypoparathyroidism, Deafness, Renal dysplasia syndrome) group, and in some individuals of other groups, a predominantly high-frequency hearing loss is seen (Figure 1). This may have consequences for the audibility of speech in the speech-in-noise test [31]. Therefore, the CDF values are corrected for the inaudibility of speech in the higher frequencies, using the simplified method previously described [32,33]. This correction is applied to individual data. For USH2a, the CDF of 8 of the 11 patients is corrected; the mean correction factor is 0.9 dB (range: 0.2–1.8 dB). The audibility-corrected CDF values are used for further analyses. For the HDR patients, corrections are low (between 0 and 0.3 dB), which were subsequently neglected. The Muckle-Wells patient group comprised only one patient and was therefore excluded with the present inclusion criteria. The DLF test was not carried out in the group of DFNA10 patients.

Table 1 presents an overview of the studies in terms of the total number of subjects, a description of the deafness [34] and its corresponding gene, its audiometric phenotype according to the GENDEAF guidelines, the main findings concerning described psychophysics, and the suggested cochlear site-of-lesion based on gene expression profiles [35] along with the type of hearing loss [18]. Further phenotypical information, such as acoustic reflexes, otoacoustic emissions (OAEs), and vestibular tests, were obtained in only a subset of the studies and were thus excluded from further analysis. 

To investigate the present research hypothesis, we decided to homogenize the nine groups of patients and to only include data of patients aged between 17 and 70 years. These inclusion criteria were previously introduced and relate to presbyacusis as a factor that might interfere with hereditary hearing impairment [22] and can thus be considered as an upper age limit. The lower age limit was introduced due to problems observed in children and adolescents while performing some of the subtests [23]. Furthermore, the degree of hearing loss was homogenized; the individual pure tone average (average hearing loss at 0.5, 1, 2, and 4 kHz, or PTA) had to be moderate to severe, between 30 dB HL and 75 dB HL.

Table 2 presents the mean age and mean PTA with their ranges for the different studied groups of patients. Figure 1 shows the mean audiogram per patient group; most audiograms are relatively flat or mildly sloping.

In retrospect, the speech recognition in noise scores is related to the loudness growth data, gap detection threshold, and difference limen for frequency, using regression analyses. Additionally, compared to the literature, the speech-in-noise data are related to the generic variables of age and hearing loss. As the outcomes of the speech-in-noise test primarily depend on the processing of high-frequency information [3], the psychophysical data obtained at 2 kHz are used (i.e., the highest of the two frequencies tested).

## 3. Results

Table 1 shows the type, severity, and progression of the hearing loss for all forms of genetic hearing impairment. It also presents a qualitative description of the results on the loudness perception task, the gap detection task, the difference limen for frequency, and the qualitative SRT, all relative to the normal values. It also shows the results from the acoustic reflex test, presence or absence of otoacoustic emissions, and vestibular test results when these were available. Finally, a description of the type of hearing loss is given [18] and the hypothesized affected cochlear site-of-lesion based on gene expression profiles [35]. 

Table 2, column 4 presents the mean (audibility corrected) CDF as calculated from the speech recognition in noise scores from the patient groups and its range, which is 2 dB or less in most patient groups, suggesting good reproducibility [11]. Audibility corrections of the CDF have been performed on individual patients. Figure 2 and Table 2 show that the CDF varies substantially between patient groups; in DFNA8/12 and DFNA13 (Figure 2, first two box plots and Table 2, rows 1 and 3), its value is relatively close to 0 (corresponding to normal hearing), which is indicative for normal speech recognition in noise. The poorest results are seen for the DFNA10 and USH2a patient groups (Figure 2, 7th and 8th box-plots and Table 2, rows 2 and 6), even after audibility corrections. 

The Cochlear Distortion Factor (Figure 3) is affected by the pure tone average but not by age.

Following the literature [4,36], the relation between speech-in-noise scores and the generic variables hearing loss (PTA) and age was also studied. The simple linear regression (ordinary least squares) of the CDF with variables PTA and age showed a good overall fit of the model (F (2,35) = 14.8, *p* < 0.001) with an r^2^ of 0.3. Interestingly, only the CDF was significantly related to the PTA; the CDF increased by 1.5 dB per each 10 dB increase of the hearing threshold (PTA). The factor age, however, is not significantly related to the CDF. Since age and PTA are strongly correlated, as we know from age-related hearing loss (ARHL), we also tested the effect of age by omitting the factor PTA in the linear model. In this model, age is also not significant. This is an important finding because cognition is highly correlated with age and is known to also affect the CDF [37,38]. So, while we did not test for differences in cognition between subjects and groups, the lack of a significant effect of age on the CDF may thus be (indirectly) suggestive of limited effects of cognition.

### 3.1. The Cochlear Distortion Factor Depends on the Specific Genes Affected

Despite the apparent relation of CDF with PTA, it can be observed that the CDF also varies with respect to the factor *‘patient group’*. For some groups, such as the Usher2A patients, most patients (i.e., 7 out of the 10 patients with a complete data set) have a CDF that falls above the regression line and its 95%-confidence interval. In another group, such as the DFNA13 group, half of the participants (4 out of 8 patients) have a CDF that falls below the regression line. This indicates that, although CDF and PTA seem related, the regression model with only PTA and age fails to capture most of the variance. Indeed, by adding the variable ‘patient group’ as a categorical variable, the model captures more explained variance from r^2^ = 0.42 (variables PTA and age) to r^2^ = 0.75 (variables ‘patient group’, PTA and age; model fit: F (9,28) = 9.53, *p* < 0.001). 

### 3.2. Introduction of Two Models

Since the variation of CDF within the patient groups is substantial, the linear regression analysis was also performed on the mean CDF data for each group and the three psychophysical variables. The linear regression analysis using the relative slope of the loudness growth curve (SLG) and the relative gap detection threshold (GDT) as variables raised the explained variance (r^2^) from 0.63 (only SLG) to 0.88 (group average model 1, equation 1: both GDT and SLG; F(2,7) = 18.6, *p* = 0.004)). Adding the difference limen for frequency (DLF) instead of gap detection also improved the r^2^ from 0.87 to 0.98 (group average model 2, equation 2: DLF and SLG; F(2,7) = 95, *p* < 0.001). These r^2^ values are very high. The final equations for the two models are: CDF = 3.6 × SLG + 2.2 × GDT − 4.9 (1)
CDF = 4.9 × SLG + 1.5 × DLF − 7.1 (2)

According to these two models, a normal-hearing subject (SLG = 1, GDT = 1, DLF = 1) has a CDF factor of 0.9 dB and −0.7 dB, respectively, for model 1 and model 2, thus being close to the expected CDF of 0 (no cochlear distortion). At a group level, the CDF could therefore be predicted by the combination of the loudness growth (SLG) and either the relative gap detection threshold (model 1) or the relative difference limen for frequency (model 2). The fits are both significant, show a high degree of explained variance, and show that the two models also hold for normal-hearing participants. A third model where all three variables were used in the linear model was also considered. While it also explained 98% of the variance, an ANOVA comparing this model with model 2, the best model, yielded a non-significant result (*p* = 0.28). The third model was thus rejected as model 2 has only two explaining variables while it explained the data equally well.

## 4. Discussion 

Using previously published data on different groups of patients with genetic hearing loss, we have shown that cochlear distortions vary considerably between patients with different types of genetic hearing loss. This variation in cochlear distortion, and thus the impairment, between the various patient groups, suggests dysfunction at specific cochlear subsites (e.g., hair cells, tectorial membrane) or site-of-lesions, where some forms are more detrimental to understanding speech-in-noise than others. This, in turn, may explain the often-reported poor relationship between hearing loss (PTA) and speech recognition. Moreover, regression models indicate that loudness growth and spectral resolution best describe the cochlear distortions and are thus a good biomarker for speech understanding in noise.

### 4.1. Classification of Hearing Loss–New Insight 

With growing knowledge in the field of genetic hearing loss and molecular biology of the inner ear, the classification system as introduced by Schuknecht and Gacek has been debated over the years [39,40]. Nevertheless, we argue that the classification system, although it has its limitations, remains a good starting point for the classification of other forms of hearing impairment, such as genetic forms of hearing loss. It may explain the heterogeneity we see in hearing thresholds based on a more fine-grained picture of the affected structures in the cochlea, as probed by speech audiometry and more advanced psychophysical tests. For example, in data where word-recognition scores were available, auditory nerve fiber (ANF) survival was predictive of word-recognition scores as an additional factor to PTA [21]. The question remains whether such a classification system will ultimately predict the ability to understand speech and speech-in-noise as there are many combinations of inner ear pathologies, each to various degrees contributing to both hearing sensitivity (i.e., thresholds of hearing) and suprathreshold measures, such as understanding speech-in-noise. Yet, with well-genotyped groups of patients, and sensitive audiological outcome measures, we may learn about specific aspects of inner ear pathology and their relationship with perception, as probed by, e.g., speech-in-noise tests.

### 4.2. Speech-in-Noise Performance and Underlying Pathology: Some Groups Do Well While Others Struggle

The first published study from our group (see Table 1, de Leenheer et al. in 2004) dealt with the effect of mutated *COL11A2* on cochlear function in patients with DFNA13. This deficient gene exhibits a loss of organization of the collagen fibrils in the tectorial membrane, thus affecting the viscoelastic properties of this membrane. Owing to the near-normal CDF and near-normal slopes of the loudness growth curves and the tectorial membrane anomaly, it was stated that the hearing impairment acted as a cochlear conductive type of hearing loss. In a second study, similar outcomes (near normal CDF, normal loudness growth) were reported in patients with DFNA8/12 [25]. These patients also have a disrupted structure of the tectorial membrane matrix due to pathogenic variants in the *TECTA* gene. Pathogenic variants in genes that affect the tectorial membrane function, such as *COLL11A2* and *TECTA*, are thus likely causing hearing loss with near-normal speech-in-noise performance and normal loudness growth. Subjects in these groups perform well when audibility is restored by using, e.g., hearing aids.

A second sub-group of pathology can be identified as a ‘sensory’ type of hearing loss, following Schuknecht and Gacek (1993) and Ohlemiller (2004). Based on the outcomes of the psychophysical tests and the present knowledge of pathology on a cellular level [35], Usher syndrome type 2a, DFNA22, DFNA10, and HDR syndrome were categorized as sensory types of cochlear hearing loss [13,24,26,27] where the hair cells are affected by specific pathogenic variants (see Table 1). DFNA22, characterized by myosin VI defects, leads to hair cell problems where the anchoring of hair cell membranes is affected. In mice models with a deletion in the MYO6 gene, the stereocilia appear normal at birth by fuse in the first days and show progressive disorganization of hair cells, finally leading to complete degeneration of both inner and outer hair cells. 

Yet, categorization of the type of cochlear hearing loss might be complicated by inter-subject variations, as was found in the outcomes of patients with the non-ocular Stickler syndrome [28]. Note, however, that if the deficient gene affects the tectorial membrane, this does not necessarily mean that the hearing loss is unique to the cochlear conductive type. Within the group of non-ocular Stickler patients, both sensory and cochlear conductive types of hearing loss seem to be present [26]. The function of the hair cells in these patients might be more negatively influenced by insufficient contact with the impaired structure of the tectorial membrane and by the changes in the elastic properties of the membrane affecting the sensitivity of (otherwise normal) hair cells [17]. Another example is the DFNA18B/DFNA84B group. In this group, pathogenic variants cause changes in Otogelin and Otogelin-like proteins that also affect the tectorial membrane. Unlike the other examples (DFNA8/12 and DFNA13), the audiological data show increased CDFs and steeper than normal loudness growth curves. A possible explanation might be that the outer hair cells, with their stereocilia in contact with the tectorial membrane, do not function normally because of an ineffective connection [41]. 

In summary, there is a group of patients that have no apparent cochlear distortion and show normal speech perception (e.g., DFNA8/12, DFNA13). In these groups, pathogenic variants have an impact on how the tectorial membrane functions. Yet, while there are several genes where pathogenic variants affect the tectorial membrane, not all cases lead to (near) normal speech perception and normal loudness growth. Instead, in some groups, it may reflect a combination of sensory and cochlear conductive hearing loss (e.g., DFNA18a/DFNA84B, non-ocular stickler). In these groups, the connection of the tectorial membrane with the OHCs also seems affected, leading to more sensory loss or mixed loss.

Lastly, the groups identified as a sensory loss, such as the groups of patients with DFNA22, Usher type 2a, HDR syndrome, and DFNA10, perform worse than normal, even when correcting for audibility. These groups show a higher CDF and thus show worse performance on speech-in-noise tests related to the underlying pathology of hair cells. 

### 4.3. Underlying Psychophysical Variables Explain Speech Understanding

In the studies reviewed and selected, the speech-in-noise performance was measured as well as the psychophysical variables, such as the slope of the loudness growth curve, the gap detection threshold, and the difference limen for frequency. These variables may predict speech-in-noise performance [6,42,43]. At a group level, speech-in-noise could be predicted by the combination of the loudness growth (SLG) and either the relative gap detection threshold (model 1) or the relative difference limen for frequency (model 2). It shows that the ability to perceive loudness differences, temporal processing of speech, and spectral abilities are essential for speech-in-noise performance in line with earlier work [42,43]. The results show that loudness perception and spectral abilities explain the data better (r^2^ = 0.98) than the loudness perception combined with temporal processing abilities (r^2^ = 0.88). A model where all variables were considered was less parsimonious, did not yield a better fit, and was therefore rejected. While speculative, the fact that the loudness perception and spectral abilities are the most critical variables but to a lesser degree, the temporal coding may indicate that the groups studied show affected cochlear processing rather than more upstream problems with, e.g., temporal coding. Notably, these analyses were performed at the group level since a substantial variation of the speech recognition in noise was observed within each group. 

### 4.4. Limitations

A limitation of the present retrospective study is the limited number of groups and the limited number of patients per group (min: 3, max: 14; after exclusion, min: 3, max: 7). Recruiting sufficient numbers of patients with a well-established genetic hearing loss diagnosis is troublesome because most types of genetic hearing loss are rare or very rare. Pooling data from different centers is thus essential to deal with the low numbers and should be part of future research. A second limitation is that, despite our efforts to limit the range of hearing loss, the patients per group were still heterogeneous regarding their PTA (i.e., with varying degrees in PTA within the groups, see Table 2 and Figure 1). In addition, genetic heterogeneity was also seen in the studied groups. The Usher2a and non–ocular Stickler groups comprised members from different families with different pathogenic causative variants. This may be important as the auditory phenotype may vary substantially with the specific variant found [44,45]. Even within families where siblings share the same pathogenic variant and environment, the phenotype may vary considerably [46], indicating that modulating variables, such as modifying genes, epigenetics, and environmental factors may cause differences. Finally, speech-in-noise is dependent on cognitive processes, such as working memory [47] and attention [38], especially with increasing age (>65). These tests were not taken into account by the original papers reviewed here, and we can only speculate on cognitive effects. Our data shows no age effect on the speech perception in noise when controlling for hearing loss. This is important as hearing loss has been considered a strong mediating factor in explaining the correlation of cognition with age, as untreated hearing loss drives the association between hearing loss and cognition [48]. The fact that age has no contribution to performance does not rule out cognitive factors but points more towards a cochlear site-of-lesion. Indeed, differences in speech understanding between groups are readily observed within the same age range (Figure 1 and Figure 2). At the same time, we do not have clear indications that the groups differ substantially in their cognitive abilities.

So, with these limitations in mind, we want to identify the results as being far from definitive. Instead, we want to allude to the idea of a site-of-lesion and cochlear processing that may be affected by pathogenic variants, either associated with anatomical or functional changes in the cochlea. This is a first approximation in understanding the variation we observe in the ability to understand speech-in-noise and relating the obtained cochlear distortion (CDF) to underlying psychophysical features. To further study cochlear processing, it is recommended to use a universal test battery, such as the ‘Auditory profile’ test battery [36,49], or more recent work on the Better Hearing Rehabilitation (BEAR) project [50]. 

A better understanding of aspects of hearing impairment, as captured by a ‘deep phenotype’ of hearing loss, may reveal essential biomarkers desperately needed in advancing treatment [51,52]. We see this as a necessary step between understanding what is happening in the cochlea at a fundamental level and how it translates to a more functional level, captured by biomarkers or auditory profiles [50,52], ultimately leading to our ability to understand speech-in-noise.

The categorization of cochlear hearing loss, e.g., cochlear conductive versus sensory, is very important for rehabilitation. Suppose the hearing loss is, for example, of the cochlear conductive type. In that case, linear amplification using hearing aids might be beneficial for patients, similar to the ‘classical’ fitting of a (true) conductive hearing loss. In contrast, in the case of outer hair cell loss, a more compressive form of amplification might be the better choice to deal with loudness recruitment. Furthermore, in the latter group, noise reduction and speech enhancement strategies might be beneficial in dealing with the broader rather than the normal auditory filters [30]. Based on our findings, we thus advise audiologists to fit hearing aids in patients with DFNA8/12 or DFNA13 with a more linear amplification program. 

Moreover, the categorization and identification of the site-of-lesion is important for patient selection for upcoming trials [51,52]. A deep auditory phenotype, possibly combined with a clear genotype when considering new forms of gene therapy, is thus essential for patient selection and for evaluation of inner ear therapeutic studies. This deep phenotyping becomes even more important when the genotype of, e.g., age-related hearing loss is yet to be established.

In summary, different types of genetic hearing loss might uniquely affect cochlear processing, resulting in different auditory profiles that can be assessed by psychophysical tests. Such knowledge might help shape the expectations of patients referred for hearing aid fitting in the clinic and provide some insight into better ways to start hearing loss rehabilitation or even provide a treatment. Furthermore, the lack of predictive power at the individual level suggests that other variables could explain more of the variance we observe within the groups. This deep phenotype of hearing loss is needed, if only to have a good tool for selecting suitable patients for new and upcoming inner ear therapeutic studies [51], as an example of precision medicine.

## Figures and Tables

**Figure 1 genes-13-01923-f001:**
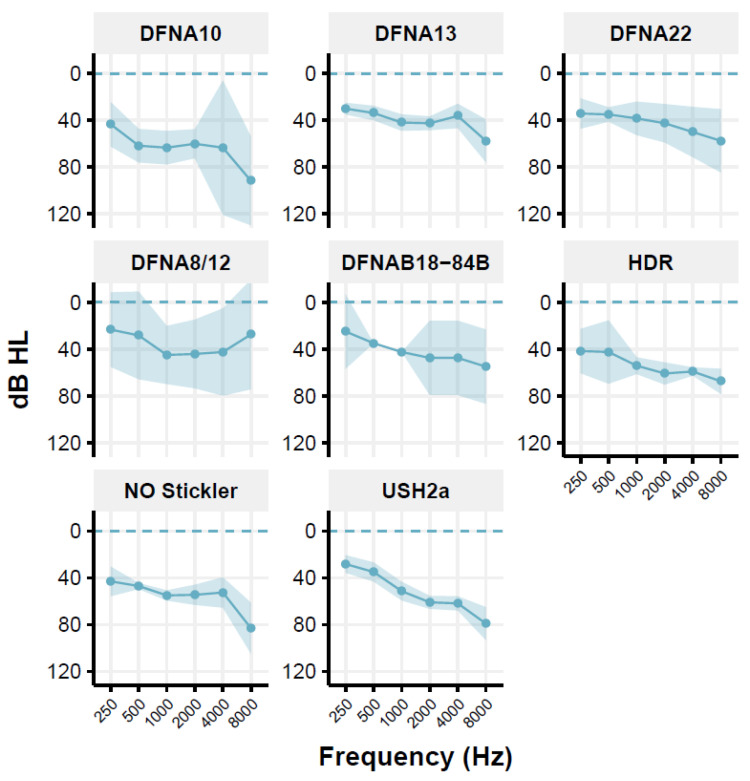
Mean audiogram of the patient groups showing the mean threshold for the frequencies tested (right and left ear averaged), as well as the 95% confidence interval around the mean.

**Figure 2 genes-13-01923-f002:**
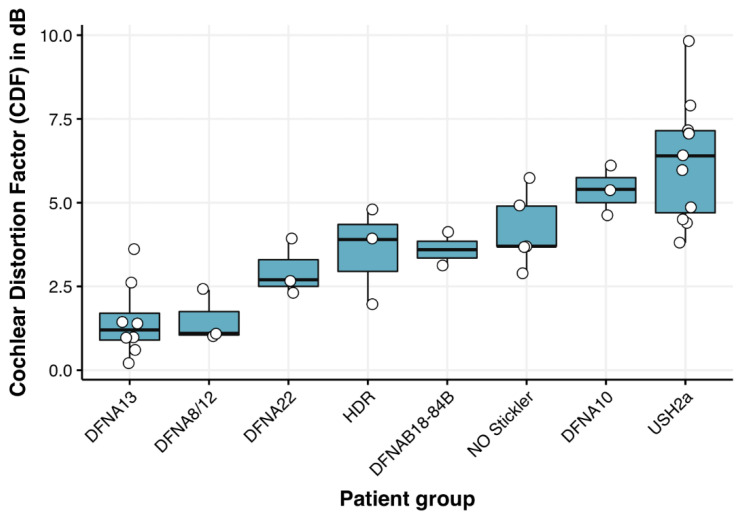
Boxplots of the eight patient groups, rank-ordered by the mean CDF for each group. The groups vary with respect to their mean CDF, but also with respect to the distribution within each group. The highest mean CDF (i.e., poorest speech understanding in noise) and the broadest distribution of individual patients’ CDF-values can be observed in the USH2a group. CDF = 0 means normal speech understanding in noise.

**Figure 3 genes-13-01923-f003:**
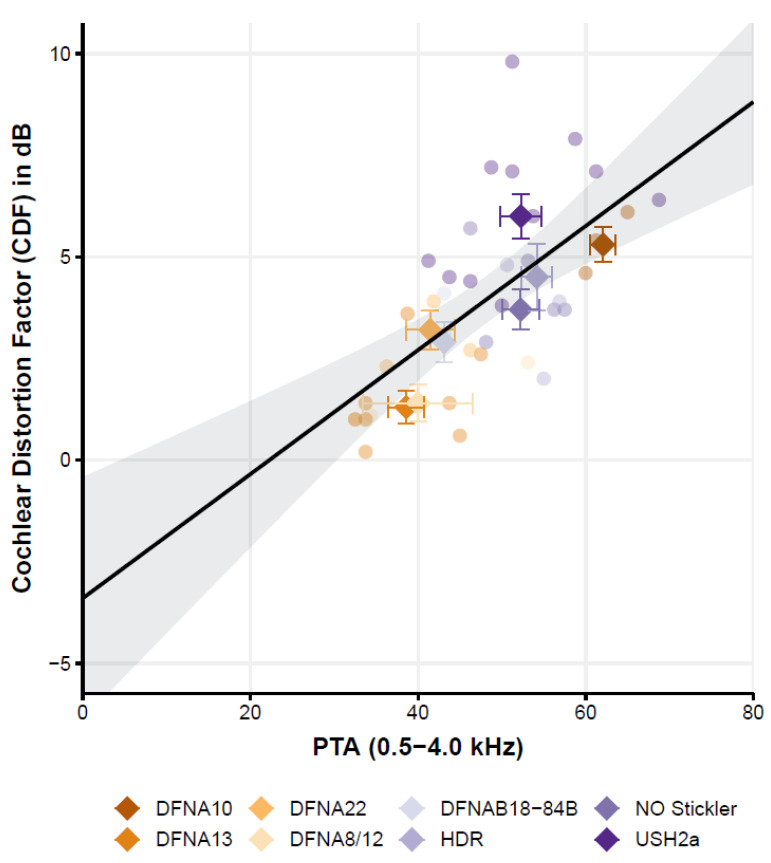
Cochlear distortion factor (CDF) as a function of the hearing loss (PTA) of the eight patient groups (individual data shown as circles where each patient group has a different color. The average for each patient group across PTA and CDF is indicated by a diamond-shaped symbol). The linear regression line presents the calculated best-fit curve and the 95% confidence interval of the fit.

**Table 1 genes-13-01923-t001:** Included and excluded patients with genetic hearing impairment taken from previous studies.

* Deafness Type *	* Gene *	* Mutation *	* Severity *	* GENDEAF *	* Progression *	* Loudness Perception *	* Acoustic Reflexes *	* Gap Detection *	* DLF *	* SRT *	* Description *	* Site-of-Lesion (Suggested or Presumed) *	* Vestibular *
**DFNA13**	*COL11A2*	*c.2423G > A (p.Gly808Glu)*	mild to moderate/severe	Early in life low-middle frequencies; later bilateral gently to steeply sloping	no	LDL higher than normal; loudness growth comparable to NH	Elevated compared to NH (esp 2–4 kHz)	normal in younger subjects	normal in younger subjects	normal in younger subjects	cochlear conductive	Tectorial membrane	na
**Non-ocular Stickler syndrome**	*COL11A2*	*c.3659G > A (p.Gly1220Asp)*	mild to moderate/severe	gently to steeply sloping high-frequency HL.	no	loudness growth comparable to NH	na	slightly increased	increased	increased but better than presbyacusis	cochlear conductive	Tectorial membrane	na
**DFNA8/12**	*TECTA*	*c.5668C > T (p.Arg1890Cys)*	mild to moderate	bilateral mid-frequency U-shaped	no	LDL higher than normal; loudness growth comparable to NH	Elevated compared to NH	except one subject normal	normal in all but one	except one subject close to normal	cochlear conductive	Tectorial membrane	na
**Muckle-Wells syndrome**	*NLRP3*	*c.2575T > C (p.Tyr859His)*	moderate to severe	gently to steeply sloping high-frequency HL.	1.3–1.8 dB/year	loudness growth steeper than NH at 2 kHz	na	normal	increased	close to normal	cochlear conductive	Basilar membrane	variable
**DFNA18B/84B**	*OTOG & OTOGL*	*c.547C > T (p.Arg183X) & c. 5238+5G > A) c.1430delT (p.Val477Glufs*25) c.5508delC (p.Ala1838ProfsX31) c.6347C > T (p.Pro2116Leu) & c.6559C > T (p.Arg2187X)*	mild to moderate	gently donwsloping	no in 3 families; 0.53–1.17 dB in two individuals	loudness growth steeper than NH at 2 kHz	normal	slightly increased	slightly increased	increased but better than presbyacusis	sensory	Tectorial membrane/connection with stereocilia	hyporeflexia & delayed motor development
**DFNA22**	*MYO6*	*c.3610C > T (pR1204W)*	mild to severe	gently to steeply sloping high-frequency HL.	Similar to presbyacusis	loudness growth steeper than NH at 2 kHz	na	normal	normal	increased but better than presbyacusis	sensory	Stereocilia of the hair cell	except one subject with hyporeflexia left normal vestibular function
**Usher syndrome type 2a**	*USH2A*	*see Table 1 of the publication for the various mutations*	moderate to severe	gently to steeply sloping high-frequency HL.	na	loudness growth steeper than NH at 2 kHz	na	increased tresholds	increased	increased; similar to prebyacusis	sensory	Hair cells bundle links	na
**HDR syndrome**	*GATA3*	*c523_528dup (p.Gln178ProfsX19)*	mild to moderate	gently donwsloping	no	loudness growth steeper than NH	na	increased tresholds	slightly increased	increased but better than presbyacusis	sensory	Hair cells (OHCs)	normal
**DFNA10**	*EYA4*	*c.464del (p.Pro155fs*)* *c.1810G > T (p.Gly604Cys)*	mild to moderate	bilateral mid-frequency U-shaped, some an additional high-frequency moderate hearing loss	0.5–1.6 dB/year	loudness growth steeper than NH	na	increased tresholds	unpublished	increased	sensory	Hair cells & stria vascularis	normal

HDR stands for hypoparathyroidism, deafness, renal dysplasia syndrome; *: First author and year of publication.

**Table 2 genes-13-01923-t002:** Characteristics of the patient groups.

Type of Genetic Hearing Loss	Age, in Yrs. (Range)	PTA (0.5–4.0 kHz) in dB HL (Range)	Audibility-Corrected Cochlear Distortion Factor (CDF) in dB (Range)
DFNA8/12	37 (27–45)	40 (33–53)	1.5 (0.9–2.4)
DFNA10	52 (31–65)	62 (60–65)	4.1 (4.6–6.0)
DFNA13	44 (34–63)	39 (33–48)	1.5 (0.6–2.6)
DFNA22	60 (53–66)	41 (36–46)	3.0 (2.5–4.1)
DFNB18B	19 (18–20)	43	3.6 (3.1–4.1)
Usher syndrome type 2a	40 (28–59)	52 (41–69)	5.6 (3.8–9.2)
Muckle Wells syndrome	21	60	6.6
Non-ocular Stickler syndrome	58 (44–68)	52 (46–58)	4.1 (2.9–5.7)
HDR syndrome	38 (25–56)	54 (51–57)	3.6 (2.0–4.8)

HDR stands for hypoparathyroidism, deafness, renal dysplasia syndrome.

## Data Availability

Data are available upon request.

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
