# Peer review of "Genetic Hearing Loss Affects Cochlear Processing"

_genes, 2022, doi:10.3390/genes13111923_

Round 1
Reviewer 1 Report
The research data of the author's team is relatively complete and the conclusion is reliable, which has certain clinical significance. The writing style is relatively standard, and the application of charts is reasonable.
The main problem is that some gene mutation sites are not completely clear. Although the phenotype summarized by the author has certain characteristics, the pathogenesis is not clear enough, which has brought some impact on the research value.
It can be published as a good summary.
Author Response
We would like to thank the reviewer for the time and questions. We have added more details concerning the specific pathogenic variants and mechanisms and added substantially to the discussion, highlighting that different variants may affect the phenotype differently. Broadly speaking, there are two main groups concerning speech understanding: a group with a cochlear conductive loss (i.e., tectorial membrane primarily affected) characterized by a good speech in noise ability and a group with mainly a sensory loss with, in general, worse than normal speech in noise scores. Some groups have mixed forms. It is clear that the groups are based on pathogenic variants within single genes. However, some forms are caused by multiple different variants (e.g., Usher 2a) within a gene, whereas others are less heterogeneously affected by a single pathogenic variant that inherits dominantly. Unfortunately, we do not have the statistical power to assess the phenotypic impact of different variants within these genes (c.f. Hartel et al. 2016).
1.
Hartel BP, Löfgren M, Huygen PLM, Guchelaar I, Lo-A-Njoe Kort N, Sadeghi AM, et al. A combination of two truncating mutations in USH2A causes more severe and progressive hearing impairment in Usher syndrome type IIa. Hear Res. 2016;339:60–8.
Reviewer 2 Report
The description of the papers used in this review was insufficient. The number of cases in each of the 9 studies is small generally. The authors neglected the GDT and DLF in their results and discussion. The results section is not organized and need to be rewritten again. The regression section is not clear, complicated and difficult to be understood. Discussion section is poor and very short
Author Response
We would like to thank the reviewer for highlighting the shortcomings. We have rewritten and reorganized the results section and the discussion. We have significantly restructured the analysis and interpretation of the linear models used to describe the relation of speech in noise with the variables gap detection, loudness perception, and spectral abilities. We have further highlighted the association with age. While not part of the original papers, we tried to compensate for the differences in age between groups to assess any differences that might relate to cognition (mediated through, e.g., hearing loss).
We have added more details concerning the specific pathogenic variants and mechanisms. We have added substantially to the discussion highlighting how the different variants may affect the phenotype and making a correlation of the phenotype and genotype based on mechanisms and site of lesion.
Reviewer 3 Report
The authors have attempted to report the effect of genetic hearing loss on cochlear processing. I recommend that the manuscript is not suitable for publishing in the current form due to the following reasons:
The manuscript is retrospective, and the results explained suffer from methodological control and rigor.
The manuscript discusses only cochlear processing, and the neural involvement in speech understanding is ignored. Central auditory processing is also equally necessary, and the poor PTA and SRT relationship cannot result from only cochlear damage.
The authors have reviewed previous work, collated it, and written the manuscript, which leads to the poor internal validity of the study. However, the authors have acknowledged the limitations. It is a significant factor affecting the study's quality and its claims.
Discussion should be further strengthened with a holistic perspective.
The manuscript should be proof-read for minor grammatical errors.
Author Response
We would like to thank the reviewer for their time and comments.
We have rewritten and reorganized the results section and the discussion. We have significantly restructured the analysis and interpretation of the linear models used to describe the relation of speech in noise with the variables gap detection, loudness perception, and spectral abilities. We have further highlighted the association with age and, while not part of the original papers, tried to compensate for the differences in age between groups to assess any differences that might relate to cognition (mediated through, e.g., hearing loss). As stated, the problem is that it is a retrospective analysis where we analyze the available data. Since cognition was not explicitly tested in any way, shape, or form, we cannot rule out cognitive effects. We have highlighted this shortcoming. However, we do not have any indication of substantial differences between the subject groups within the same age range. With that assumption, the speech in noise performance is different between the groups, so while we cannot rule out cognitive aspects, we think cochlear damage or dysfunction leads to poor spectral resolutions, altered loudness perception, and issues with temporal processing. We even highlight which of the parameters is most likely the main contributor (e.g., loudness perception and spectral resolution). In line with previous research, given the problems that arise in the cochlea and knowledge about the underlying etiology and mechanisms, we think a substantial part is actually due to distorted cochlear processing.
We have highlighted the shortcomings and pitfalls, presented the data fairly, and did not make strong claims. Instead, we want to allude to the idea of a site-of-lesion and cochlear processing that may be affected by genetic mutations associated with anatomical or functional changes. This is, therefore, a first approximation in understanding the variation we observe in the ability to understand speech in noise and relating the obtained cochlear distortion (CDF) to underlying psychophysical features while noting that other variables also need to be taken into account in prospective studies (i.e., a more holistic approach).
Round 2
Reviewer 2 Report
Thanks for your revision
Reviewer 3 Report
The authors have incorporated the suggested revisions and hence it can be accepted.